# Is It Possible to Have Home E-Monitoring of Pulmonary Function in Our Patients with Duchenne Muscular Dystrophy in the COVID-19 Pandemic?—A One Center Pilot Study

**DOI:** 10.3390/ijerph18178967

**Published:** 2021-08-26

**Authors:** Eliza Wasilewska, Agnieszka Sobierajska-Rek, Sylwia Małgorzewicz, Mateusz Soliński, Dominika Szalewska, Ewa Jassem

**Affiliations:** 1Department of Allergology and Pulmonology, Medical University of Gdańsk, 80-210 Gdańsk, Poland; ejassem@gumed.edu.pl; 2Department of Rehabilitation Medicine, Medical University of Gdansk, 80-210 Gdańsk, Poland; sobierajska@gumed.edu.pl (A.S.-R.); dominika.szalewska@gumed.edu.pl (D.S.); 3Department of Clinical Nutrition, Medical University of Gdansk, 80-210 Gdańsk, Poland; sylwiam@gumed.edu.pl; 4Faculty of Physics, Warsaw University of Technology, 00-661 Warszawa, Poland; mateusz.solinski.dokt@pw.edu.pl

**Keywords:** digital health, home monitoring, e-monitoring of pulmonary function, home monitoring pulmonary function, rare diseases, Duchenne muscular dystrophy, pulmonary function test, spirometry, AioCare, COVID-19

## Abstract

Background: Duchenne muscular dystrophy (DMD) is the most common, progressive, irreversible muscular dystrophy. Pulmonary function is crucial for duration of life in this disease. Currently, the European Respiratory Society is focused on digital health, seeking innovations that will be realistic for digital respiratory medicine to support professionals and patients during the COVID-19 pandemic. Aims: The aim of this study was to investigate whether it is possible to monitor pulmonary function at home using an individual electronic spirometry system in boys with Duchenne muscular dystrophy. Materials and methods: In this observational, prospective study, conducted from March 2021 to June 2021, twenty boys with DMD (aged 8–16) were enrolled. The patients were recruited from the Rare Disease Centre, University Clinical Centre, of Gdańsk, Poland. Medical history and anthropometric data were collected, and spirometry (Jaeger, Germany) was performed in all patients at the start of the study. Each patient received an electronic individual spirometer (AioCare) and was asked to perform spirometry on their own every day, morning and evening, at home for a period of 4 weeks. The number of measurements, correctness of performing measurements, forced vital capacity (FVC), forced expiratory volume in 1 s (FEV1), and peak expiratory flow (PEF) were evaluated. Results: Finally, 14 out of 20 boys enrolled in the study with a mean age of 12.5 years (7 non-ambulatory) applied and received a home spirometer (AioCare). A total of 283 measurements were performed by all patients at home for 4 weeks. Half of the patients were able to perform measurements correctly. There were no significant differences between mean values of FVC, FE1, PEF between home and hospital spirometry (*p* > 0.05) expect PEF pv% (*p* < 0.00046). Patients with higher FEV1 (*p* = 0.0387) and lower BMI (*p* = 0.0494) were more likely to take home spirometer measurements. The mean general satisfaction rating of home-spirometry was 4.33/5 (SD 0.78), the mean intelligibility rating was 4.83/5 (SD 0.58). Reasons for irregular measurements were: forgetting (43%), lack of motivation (29%), difficulty (14%), lack of time (14%). Conclusion: Home electronic monitoring of pulmonary function in patients with DMD is possible to implement in daily routines at home. This protocol should be introduced as early as possible in patients 7–8 years old with good, preserved lung function. Patients accept this form of medical care but require more education about the benefits of e-monitoring. There is a need to implement a system to remind patients of the use of electronic medical devices at home, e.g., via SMS (short message service).

## 1. Introduction

Neuromuscular disorders include a heterogeneous group of diseases that impair the function of muscles, motor neurons, peripheral nerves, and neuromuscular junctions. Duchenne muscular dystrophy (DMD) is the most common, progressive, irreversible muscular dystrophy, caused by mutations in the X-linked dystrophin gene. The function of the respiratory system in this disease is crucial for duration of life [1,2]. Genetically determined and progressive with age, the atrophy of the dystrophin protein in the muscles also includes respiratory muscles.

Many studies confirm that from the age of 7 years, lung function in children with DMD does not increase as in healthy peers. Moreover, when the child loses the ability to walk independently (usually between 10 and 13 years of age), values of pulmonary function tests rapidly deteriorate [3,4]. Basic care at this stage includes the initiation of early monitoring of respiratory system functions. It is recommended that measurement of pulmonary function is started from the age of 5 years [2,5]. According to the standards, pulmonary function is assessed by spirometry, which should be performed at least once a year, at least every 6 months after the loss of independent walking function, and every 3 months after starting non-invasive ventilation (NIV) [2,5].

The spirometry test usually assesses forced expiratory volume in 1 s (FEV1), peak expiratory flow (PEF), and forced vital capacity (FVC). The latter parameter is considered a marker of disease progression in patients with DMD, better than FEV1 or PEF, and is more useful for diagnosing obstructive diseases of the lungs. An FVC value below 2.1 L is a rationale to start supporting the cough reflex, and below 1 L is an indication to start respiratory support, i.e., NIV [6,7,8,9].

Implementation of the above-mentioned tests often encounters difficulties, which increase when the child loses independent walking function. Additionally, during the COVID-19 pandemic, spirometry was included in the procedures generating aerosols, i.e., high-risk of SARS-CoV-2 virus transmission [10,11,12]. Therefore, even before the pandemic and today, and especially during the COVID-19 pandemic, the European Respiratory Society (ERS) has focused on digital health, seeking to define the innovations that are realistic for digital respiratory medicine [13,14].

Since DMD patients are at risk of a severe course of COVID-19 [15], it is recommended, necessary, and important to seek the safest way to diagnose and monitor pulmonary function in patients with DMD. The purpose of this study was to investigate whether it is possible to monitor pulmonary function at home using an electronic spirometry system called AioCare [16] in boys with DMD. The second aim of the study was to assess the acceptance of electronic home monitoring of a respiratory system in this group of patients.

## 2. Materials and Methods

### 2.1. Study Design

This observational prospective study was conducted from March 2021 to June 2021 as the second part of the project: “E-monitoring of the pulmonary function in patients with Duchenne muscular dystrophy undergoing respiratory rehabilitation at home”; details of the first part were described elsewhere [17]. 

The patients with DMD were recruited from Rare Disease Centre (RDC), University Clinical Centre, Medical University of Gdańsk, Poland, where they are under the care of a multi-specialized team. The University Clinical Centre is a member of the TREAT NMD Alliance Neuromuscular Network. Approval for the study was obtained from The Local Committee of Ethics no NKBBN/260/2021, which conformed to the principles embodied in the Declaration of Helsinki. Written informed consent was obtained from the participants and parents of each patient.

### 2.2. Participants 

The study population included boys with DMD diagnosis based on current guidelines, the presence of typical clinical symptoms, genetic testing, and/or muscle biopsy results [1]. Inclusion criteria were as follows: (1) Male, ≥8 years and <18 years of age at time of enrollment in the study; (2) ability to perform spirometry; and (3) stated willingness to comply with all study procedures and availability for the duration of the study (4) positive PCR-COVID test performed 24–48 h before the hospital visit.

All of participants presented a negative PCR-COVID test 24–48 h before the first study visit. Caregivers of children declared that children had not had a COVID infection so far. During the first visit, medical history data and clinical symptoms of the respiratory system were collected from an interview. In addition, a physical examination, Vignos scale (VS) assessment, Brooke scale (BS) assessment, and spirometry (Jaeger, Hoechberg, Germany) were performed in all patients. Then, each patient received an electronic individual spirometer called AioCare (HealthUp, Warsaw, Poland) and was instructed together with parents how to use it during a 2-h training session. Each participant was then asked to perform on their own three correct spirometry measurements twice daily (morning and evening) at home for 28 consecutive days.

At the follow-up visit after 4 weeks, compliance, satisfaction, and problems with the use of the electronic spirometer at home were discussed. Patients completed a survey that included questions about home spirometry (see Section 2.4).

### 2.3. Pulmonary Function Tests

#### 2.3.1. Hospital Spirometry 

During the enrollment visit, spirometry at the hospital was performed using the calibrated, computerized spirometer Pneumo Screen (Jaeger, Germany) according to the European Respiratory Society and American Thoracic Society recommendations, by a certificated, experienced pediatric pulmonologist [5,18]. A minimum of three and up to five maneuvers with maximum effort were attempted by each subject. The highest value of forced vital capacity (FVC), forced expiratory volume in 1 s (FEV1), peak expiratory flow (PEF) expressed as liters (L) and liters per minute (L/min) accordingly, and percent predicted value (%pv) from correct acceptable attempts were evaluated. The results were compared with results from home electronic spirometry.

#### 2.3.2. Home Electronic Spirometry 

AioCare spirometer (AioCare^®^ spirometers, Healthup, Poland) is a small, convenient device that can be used anywhere in children (over 5 years of age) and adults. AioCare (a portable spirometer) together with the dedicated smartphone applications (for patients and physicians) and an online panel make up the AioCare System [15]. Communication between the AioCare spirometer and the application takes place via a Bluetooth 4.0 (BT LE) connection. The spirometer measures airflow using a handheld hardware module that contains the thermal micro-electro-mechanical system (MEMS)-based flow sensor and electronics. The device measures all commonly used spirometry parameters, including FVC, FEV1, and PEF and complies with the newest ATS/ERS 2019 standardization.

In the study, patients inhaled and exhaled forcefully through a mouthpiece with antibacterial and antiviral filters. The test results were sent from the AioCare spirometer via the AioCare application for iOS and Android (as used by all current smartphones). The spirometry results were available to the practitioner in real-time in the AioCare Doctor panel. 

Every time a measurement was performed, it was classified by an algorithm built into the AioCare VR software as either correct (green dot) or incorrect (red dot) based on American Thoracic Society and European Respiratory Society (ATS/ERS) criteria, which ensured good quality and repeatable measurements.

The number of measurements by each patient and highest value of FVC, FEV1, PEF expressed as liters and liters per minute, and %pv were evaluated. Correctness of the measurements was assessed as the number of patients performing measurements correctly and the proportion of measurements performed correctly in each patient. 

### 2.4. Satisfaction Survey

Details of patient satisfaction and the possibility of home monitoring were collected from a survey. Patients were asked to express their general satisfaction and the intelligibility of the instructions from the AioCare home spirometry on a 5-point scale, where “1” meant the worst and “5” meant the best score. Participants who were not able to perform spirometric measurements regularly were also asked the reason why and what would help them to take measurements more regularly. The last question concerned the benefits of home spirometry. The full survey is presented in Appendix A.

### 2.5. Stage of Disease (Vignos Scale, Brooke Scale)

The functional status of the participants was assessed by the Vignos scale (VS). The scores on the VS range from 1 to 10 (1—the subject can walk and climb stairs without assistance, 10—the subject is confined to a bed). The VS allows staging of the disease and focuses on functional ambulatory activities. This is the main scale used for characterizing the progression of disease [19]. Upper limb functional status was assessed with the 6-point Brooke scale (BS) (1—the subject can abduct their arms in a full circle until they touch above their head, 6—the subject has no useful function of the hands) [20].

### 2.6. Statistical Analysis

The results of the statistical analysis were expressed as mean ± standard deviation (SD) or median and interquartile range (IQR). The comparison analysis of the results obtained from ambulatory spirometry and results from home spirometry using the AioCare spirometer was performed using paired, parametric Student *t*-test, or non-parametric Wilcoxon signed-rank test (depending on whether compared values were normally distributed, which was evaluated using the Shapiro–Wilk test) with a significance level of *p* = 0.05.

The relationship between the compliance of spirometry parameters, such as number of days with performed spirometry tests and clinical parameters (age, BMI, AS, VS, BS) and spirometry results (FEV1%pv or FVC%pv), were evaluated using linear regression analysis. The goodness of fit of the model, based on Akaike’s information criterion (AIC), was compared between the four models. The number of independent variables was reduced using backward, stepwise regression models. This method allows for a decrease in the variables from the model that do not contribute to the explanation of the variability of dependent variables, which reduces the complexity of the model (which was necessary due to the limited sample size). The correlation between continuous variables was evaluated using Pearson correlation coefficients. 

## 3. Results 

### 3.1. Participants

Finally, 14 out of 20 boys enrolled in the study applied and received a home spirometer during visit 1. The mean age of the study group patients was 12.5 years [8,9,10,11,12,13,14,15,16], with a median VS of 5.5 (IQR 7.0) and a median BS of 4.5 (IQR 4.0). Half of the patients were non-ambulatory, and all patients were treated with steroids. The clinical data of all participants are presented in Table 1. 

### 3.2. Pulmonary Function Test

#### 3.2.1. Home Spirometry Frequency and Correctness

A total of 283 measurements, both acceptable and incorrect, were performed by all participants at home for 4 weeks (Figure 1), 44 of which (15.5%) were acceptable.

Participants measured pulmonary function for an average of 14 days with an average of 20 measurements during the study period. Of the 14 participants, 4 (28.5%) adhered to the requirement for using the device daily throughout the study (28 ± 3 days) and 2 (14%) took measurements on less than 10 days.

Half of the patients were able to perform at least one maneuver correctly, acceptable in accordance with European Respiratory Society guidelines. Details of measurements for individual patients of home spirometry are presented in Table 2. 

There was a modest and positive correlation between number of days of measurements and FEV1pv% (r = 0.60; *p* = 0.024) (Figure 2). There were no correlations between correctness and anthropometric or spirometry parameters.

#### 3.2.2. Hospital vs. Home Spirometry

The results of pulmonary function tests performed using the hospital (Jaeger) and home (AioCare) spirometer for individual patients are presented in Table 2 and the mean for the study group in Table 3. There were no significant differences between the mean values of FVC, FE1, PEF in L (L/min), or pv% between home and hospital spirometry (*p* > 0.05) expect PEF pv% (*p* < 0.00046), which was higher for ambulatory spirometry (Table 4). Moderate-to-high correlation between values of home and hospital spirometry was observed (Table 4).

#### 3.2.3. Regression Analysis

The summary of the regression analysis is shown in Table 5. Model 1 indicates better spirometry adherence in patients with a larger FEV1%pv (*p* = 0.0387). In model 2, patients with a lower BMI presented better spirometry adherence (number of days with measurements *p* = 0.0494).

Regression models showed no significant relation between frequency or correctness of home spirometry measurements and ambulatory status, VS, or BS. 

### 3.3. Survey Analysis

A survey questionnaire was completed by 12 participants (85.7%). The ratings varied from five to three. The mean general satisfaction rating of home spirometry was 4.33/5 (SD 0.78), and the mean intelligibility rating was 4.83/5 (SD 0.58). Reasons for irregular measurements reported by participants are presented in Figure 3.

The majority of respondents declared that sending reminders about measurements via short message service (SMS) would be most helpful for implementing the assessments into their daily routine.

Benefits from home spirometry were visible for 10 respondents. For four participants, the most important benefit was breathing improvement, four thought that they would perform spirometry in the hospital easier, one respondent reported that he felt more confident at home, and one thought that home spirometry was a good respiratory exercise. 

## 4. Discussion

In this study, the possibility of implementing routine electronic monitoring of pulmonary function at home in DMD patients was evaluated. All participants were of Caucasian race and treated with steroids. Half of them were non-ambulatory and used a wheelchair. The most important findings were that 70% of patients took home spirometry measurements, almost 50% of patients were able to perform measurements of acceptable quality, and this kind of e-health care was acceptable by patients with chronic disease. 

Innovative forms of lung function monitoring at home are extremely important during the COVID-19 pandemic, especially for patients with chronic diseases and dysfunction of the respiratory system. Many studies showed that respiratory function declines at a rate of 6–11% annually in patients with DMD [3,4,5]. Moreover, respiratory muscle weakness leads to secondary changes, such as decreased lung compliance, ineffective cough with deterioration of airway clearance, and repeated infections [3,4], which causes DMD patients to have a higher risk of a severe course of COVID-19 [7]. 

Therefore, measurement of pulmonary function is recommended as a key element of DMD patient care [1,3]. Until now, spirometry procedures have been offered in special, dedicated institutions, mainly hospital or outpatients departments. This method of examination seems to be particularly burdensome for patients living far from specialized centers or with an advanced degree of disease. In addition, during the pandemic, spirometry was considered to be a procedure generating aerosols with an increased risk of SARS-CoV-2 transmission, so in many countries, it was completely banned or restricted during the pandemic [21]. 

Therefore, we tested a home spirometer with the ability to automatically transfer results in real-time to a doctor’s panel. We opted for the AioCare system because it had been tested at home on large groups of patients with other lung diseases, such as asthma [22,23], and as a screening tool for early detection of chronic obstructive pulmonary disease in primary care in both adult [24] and pediatric populations [25].

In our study, we showed that almost a third of our patients with DMD performed daily measurements of spirometry at home, whereas in another study with asthmatics, only 13% of patients succeeded every single day over the three-week study period [22]. Interestingly, patients with higher values of pulmonary function tests and lower BMI were more likely to take daily home spirometry measurements. This may indicate that patients with preserved lung function are more likely to perform dynamic breathing maneuvers. Therefore, it may be appropriate to start home spirometry in DMD patients at the earliest possible age from which patients can perform the correct breathing maneuvers, i.e., in children from 7 to 8 years old. This is also confirmed by information obtained by telephone from the mother of one of our adolescent patients, who gave up measurements after only 3 days due to rebellion before subsequent tests reminding him of the disease. Problems with compliance with the adolescent patient are known [26].

On the other hand, although all our patients declared their desire to monitor pulmonary function at home, 30% of them took less than 10 measurements during the study period. Patients with asthma were slightly more likely to take measurements; over a 3-week study, 86% of patients performed spirometry measurements at least three times within 7 days each week of the study [22].

We noticed that patients are not accustomed to innovative forms of digital medicine, including regular self-monitoring of lung function at home, moreover, with the possibility of a doctor simultaneously previewing the results. This is confirmed by the fact that the majority of our patients declared that sending a reminder about measurements via SMS would be most helpful for implementing the assessments into their daily routine. Other researchers had similar conclusions, noting that patients wanted to be reminded by daily text message about home spirometry [22].

The general satisfaction and acceptance of home spirometry was high in our patients. Interestingly, in addition to forgetting about measurements, as many as almost 30% of patients reported no motivation to take measurements. Other patients noticed benefits of home spirometry such as improving breathing maneuvers and reducing fear of spirometry in the hospital. This may lead to the fact that, in order to motivate patients more to cooperate, more attention should also be paid to the information and educational aspect of the benefits of home spirometry. 

A limitation of the study is the relatively small group of subjects, short study period, and also selection bias (only 14/20 finished the study). The study participants were exclusively white boys, and all participants were treated with oral steroids. However, on the other hand, this is a pilot study with an innovative digital form of monitoring pulmonary function at home, without the risk of COVID infection, which can support the safest way to provide constant care of DMD patients. Therefore, particularly during the pandemic period, this is an alternative and interesting proposition and safer than conventional spirometry in medical units. It has the potential to change everyday practice, drawing clinicians’ attention to the possibility of using home spirometry e-monitoring by sending data in real-time to a doctor’s panel. 

To the best of our knowledge, this is the first study concerning digital medicine devices in monitoring pulmonary function in DMD boys. To our knowledge, there is only one study on patients with DMD with home monitoring of pulmonary function, which included a simple device that only measures PEF and without the ability to send data in real-time. As a result, medical staff were obliged to call or personally visit patients to check whether patients took their assessments. The data recorded on the device were downloaded via USB connection and transferred to the study database during the study site visit [27]. Such procedures do not meet the criteria of digital medicine and do not provide security in a pandemic.

In summary, today, during a time of social distancing, seeking modern means of communication, treatment, and monitoring of patients has become an important need. Telemedicine offers many possibilities and includes a growing variety of applications and services that use telephone lines, videos, e-mails, smartphones, wireless tools, and other forms of telecommunication technology [28], which should be an interesting proposition not only for practitioners but also for our young patients. This study shows the feasibility, challenges, and acceptance of home electronic monitoring of pulmonary function in patients with DMD, indicating that such way of care is acceptable and possible to implement into daily routines at home.

## 5. Conclusions

Home electronic monitoring of pulmonary function in patients with DMD is possible to implement into daily routines at home. This protocol should be introduced as early as possible in patients 7–8 years old with good, preserved lung function. Patients accept this form of medical care; however, they require more education about the benefits of e-monitoring. There is a need to implement the system to remind patients of the use of electronic medical devices at home, e.g., via short message service (SMS).

## Figures and Tables

**Figure 1 ijerph-18-08967-f001:**
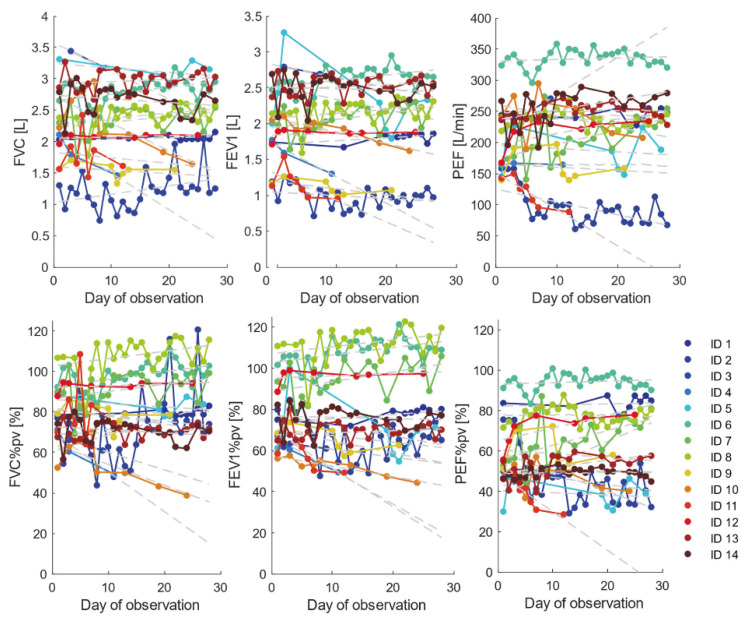
Pulmonary function test. Number of days of measurements of home spirometry for each individual patient.

**Figure 2 ijerph-18-08967-f002:**
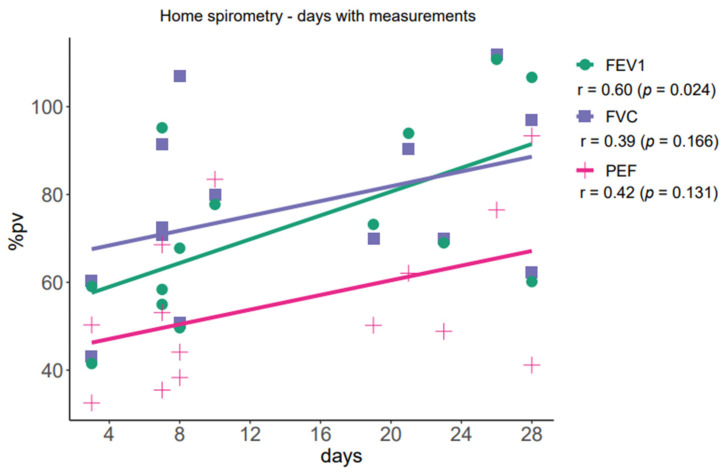
Correlation between number of days of measurements and pulmonary function tests. FVC, forced vital capacity; FEV1, forced expiratory volume in 1 s; PEF, peak expiratory flow.

**Figure 3 ijerph-18-08967-f003:**
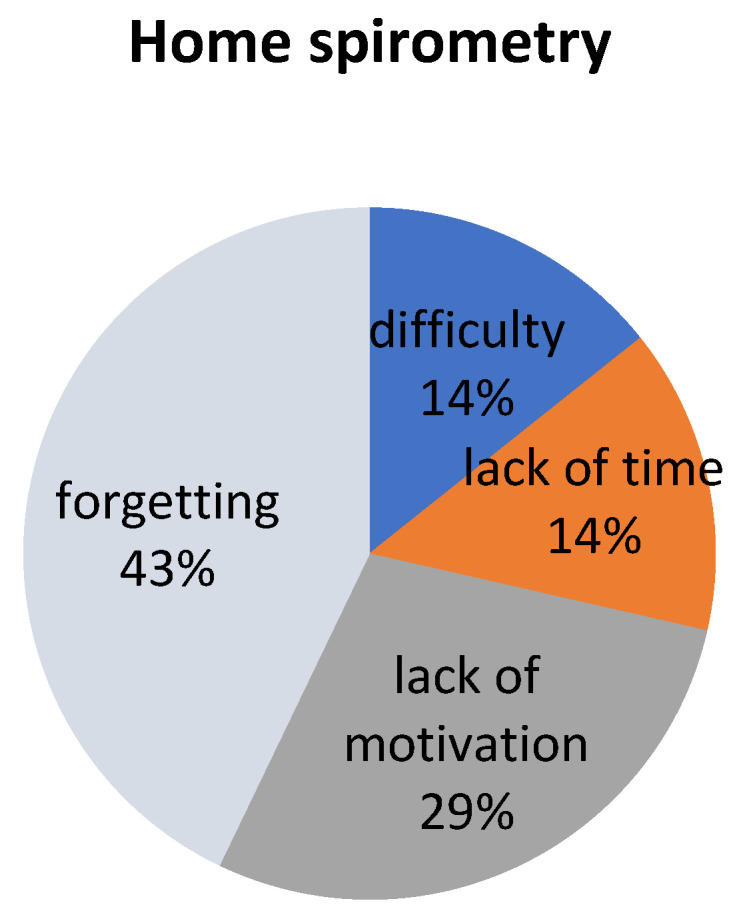
Reasons for irregular measurements of home spirometry reported by participants.

**Table 1 ijerph-18-08967-t001:** Anthropometric characteristics of patients.

ID	Age	Weight		Height		BMI		AS	VS	BS
	Years	kg	Centile	cm	Centile		Centile			
1	13	34	3	137	1	18.1	40	1	2	6
2	10	24.5	1	119	1	17.3	48	1	2	3
3	14	57	52	145	1	27.1	96	1	3	4
4	15	90	97	181	80	27.5	97	0	9	4
5	15	82	97	160	1	32.0	99	0	8	5
6	11	55.5	88	150.5	40	24.5	94	1	1	6
7	10	42	76	142	38	20.8	86	0	8	6
8	10	50	93	140	34	25.5	97	0	8	6
9	11	48	76	133	16	27.1	91	1	2	5
10	16	64	79	170	64	22.1	81	0	9	1
11	10	56	95	141	28	28.2	98	0	9	5
12	9	43	88	138	41	22.6	93	1	1	2
13	16	49	8	160	2	19.1	34	1	1	1
14	15	70	70	166	7	25.4	91	0	9	2
Mean ± SD/Median (IQR)	12.5 ± 2.6	54.7 ± 17.6	65.9 ± 35.7	148.9 ± 16.7	25.3 ± 25.6	24.1 ± 4.3	81.8 ± 23.0	-	4.9 ± 3.4	4.0 ± 1.9

BMI, body mass index; AS, ambulatory status: non-ambulatory = 0, ambulatory = 1; BS, Brooke scale; VS, Vignos scale; BS, VS are described in Section 2.5.

**Table 2 ijerph-18-08967-t002:** The number of measurements of home spirometry for individual patients.

Home Spirometry					
ID	Days of Measurements		Total Measurements	Acceptable Measurements	
	nb	%	nb	nb	%
1	10.0	35.7	11	8	73
2	28.0	100.0	57	0	0
3	3.0	10.7	3	0	0
4	3.0	10.7	3	0	0
5	8.0	28.6	11	0	0
6	28.0	100.0	45	0	0
7	21.0	75.0	27	0	0
8	26.0	92.8	28	21	75
9	7.0	25.0	12	3	25
10	8.0	28.6	12	2	17
11	7.0	25.0	10	0	0
12	7.0	25.0	8	0	0
13	19.0	67.9	27	5	19
14	23.0	82.1	29	5	17
Mean ± SD	14.1 ± 9.5	50.5 ± 33.8	20.2 ± 16.0	3.1 ± 5.7	16.1 ± 26.1

Nb, number.

**Table 3 ijerph-18-08967-t003:** Individual patient pulmonary function tests results.

ID	Home (AioCare) Spirometry						Hospital (Jaeger) Spirometry					
	FVC (L)	FEV1 (L)	PEF (L/min)	FVC (%)	FEV1 (%)	PEF (%)	FVC (L)	FEV1 (L)	PEF (L/min)	FVC (%)	FEV1 (%)	PEF (%)
1	2.07 ± 0.05	1.80 ± 0.07	243 ± 10	79.9 ± 1.7	77.7 ± 2.8	83.5 ± 3.6	1.73 ± 0.22	1.44 ± 0.71	208 ± 1.33	77.4 ± 0.43	76.9 ± 0.41	80.7 ± 0.45
2	1.05 ± 0.30	0.90 ± 0.19	86 ± 27	62.3 ± 17.8	60.2 ± 12.6	41.2 ± 12.9	1.22 ± 0.42	1.2 ± 0.45	98 ± 6.55	87.1 ± 2.25	98.7 ± 1.44	52.8 ± 4.31
3	1.79 ± 0.33	1.55 ± 0.24	165 ± 2	60.4 ± 11.0	59.1 ± 9.0	50.3 ± 0.5	2.22 ± 0.32	1.79 ± 0.43	244 ± 2.2	84.1 ± 0.41	81.2 ± 0.23	83.0 ± 0.33
4	2.25 ± 1.84	1.83 ± 1.58	170 ± 144	43.2 ± 35.3	41.5 ± 35.8	32.5 ± 27.7	2.94 ± 0.34	2.63 ± 0.9	333 ± 17	65.5 ± 0.34	69.7 ± 2.3	69.6 ± 3.5
5	4.02 ± 1.97	2.22 ± 0.52	186 ± 43	106.9 ± 52.3	67.8 ± 15.9	38.3 ± 8.9	2.82 ± 1.22	2.09 ± 0.21	212 ± 14	87.5 ± 0.93	76.5 ± 1.32	57.3 ± 2.11
6	2.78 ± 0.19	2.59 ± 0.15	332 ± 14	97.0 ± 6.5	106.7 ± 6.4	93.4 ± 3.8	2.76 ± 0.55	2.49 ± 0.65	321 ± 11	101.8 ± 3.21	107.7 ± 2.54	100.1 ± 1.54
7	2.32 ± 0.23	2.05 ± 0.20	195 ± 35	90.4 ± 9.0	93.9 ± 9.0	62.1 ± 11.2	2.13 ± 0.34	1.92 ± 0.31	231 ± 11	85.7 ± 0.34	92.1 ± 2.11	82.6 ± 2.57
8	2.49 ± 0.61	2.14 ± 0.18	233 ± 19	111.8 ± 27.4	110.8 ± 9.1	76.5 ± 6.2	2.13 ± −0.43	1.86 ± 0.54	232 ± 14	89.4 ± 2.12	93.1 ± 3.20	85.5 ± 3.5
9	1.40 ± 0.42	1.00 ± 0.31	145 ± 49	70.8 ± 21.1	58.4 ± 17.9	53.1 ± 17.8	1.58 ± 0.55	1.15 ± 0.45	215 ± 12	77.1 ± 1.32	66.9 ± 2.54	89.6 ± 6.23
10	2.14 ± 0.48	1.81 ± 0.22	228 ± 53	50.7 ± 11.3	49.7 ± 6.0	44.1 ± 10.3	2.00 ± 0.41	1.98 ± 0.92	233 ± 16	47.5 ± 2.40	56.8 ± 3.13	54.6 ± 3.22
11	1.62 ± 0.42	1.06 ± 0.25	110 ± 26	72.5 ± 19.1	55.0 ± 12.7	35.5 ± 8.4	1.38 ± 0.74	1.13 ± 0.34	115 ± 14	56.6 ± 2.43	55.5 ± 1.34	41.6 ± 3.44
12	2.04 ± 0.10	1.84 ± 0.09	206 ± 32	91.4 ± 4.7	95.2 ± 4.5	68.6 ± 10.6	1.94 ± 0.75	1.70 ± 0.39	187 ± 31	85.0 ± 2.4	88.8 ± 4.1	71.6 ± 3.5
13	2.63 ± 0.24	2.39 ± 0.27	243 ± 41	69.9 ± 6.3	73.2 ± 8.3	50.2 ± 8.5	2.74 ± 0.34	2.48 ± 0.49	258 ± 23	77.6 ± 2,4	84.5 ± 5.2	69.8 ± 5.43
14	2.96 ± 0.14	2.52 ± 0.12	249 ± 13	70.0 ± 3.4	69.0 ± 3.3	48.8 ± 2.6	2.76 ± 0.45	2.39 ± 0.34	238 ± 32	70.0 ± 3.2	73.3 ± 1.55	59.0 ± 3.56

pv, predicted value; FVC, forced vital capacity; FEV1, forced expiratory volume in 1 s; PEF, peak expiratory flow; L, liters; %, percentage of predicted value. The value of home spirometry is presented as mean ± SD.

**Table 4 ijerph-18-08967-t004:** Difference between home (AioCare) vs. hospital (Jaeger) spirometry for the entire study group.

	Home (AioCare) Spirometry %pv	Hospital (Jaeger) Spirometry %pv	Mean Difference Home vs. Hospital Spirometry	*p*-Value *	Correlation r **
FVC (L)	2.25 ± 0.73	2.17 ± 0.57	0.09 ± 0.44	0.476	0.80
FEV1 (L)	1.84 ± 0.55	1.88 ± 0.51	−0.04 ± 0.29	0.624	0.85
PEF (L/min)	199 ± 63	223 ± 64	−23.86 ± 51.32	0.106	0.67
FVC%pv	76.94 ± 20.36	78.02 ± 14.21	−1.08 ± 15.17	0.795	0.67
FEV1%pv	72.72 ± 21.47	80.12 ± 15.23	−7.40 ± 14.43	0.077	0.74
PEF%pv	55.57 ± 18.68	71.27 ± 16.58	−15.70 ± 12.47	0.0004	0.76

* Student *t*- or Wilcoxon test; ** Pearson correlation coefficient r; pv, predicted value; FVC, forced vital capacity; FEV1, forced expiratory volume in 1 s; PEF, peak expiratory flow.

**Table 5 ijerph-18-08967-t005:** Summary of stepwise linear regression models for predicting spirometry adherence (days with performed spirometry) with the clinical, anthropometric, and spirometry data.

**Model 1 Age, BMI, AS, VS, BS, FEV1%pv**		
**AIC:**	**101.00**	
Variables:	Coefficient (95%CI)	*p*-value
Intercept	14.69 (−15.64–45.03)	0.3628
BMI	−0.73 (−1.71–0.26)	0.1744
FEV1%pv	0.234 (0.039–0.430)	0.0387
**Model 2: age, BMI, AS, VS, BS, FVC%pv**		
**AIC:**	**102.19**	
Variables	Coefficient (95%CI)	*p*-value
Intercept	24.71 (−2.81–52.22)	0.1062
BMI	−1.45 (−2.17–−0.13)	0.0494
FVC%pv	0.2226 (0.0090–0.4362)	0.0659

BMI, body mass index; AS, ambulatory status: non-ambulatory = 0, ambulatory = 1; BS, Brooke scale; VS, Vignos scale; BS, VS are described in Section 2.5; FVC, forced vital capacity; FEV1, forced expiratory volume in 1 s.

## Data Availability

The data presented in this study are available on request from the corresponding author. Full data are not publicly available due to privacy restrictions.

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
