# Peer review of "Is It Possible to Have Home E-Monitoring of Pulmonary Function in Our Patients with Duchenne Muscular Dystrophy in the COVID-19 Pandemic?—A One Center Pilot Study"

_ijerph, 2021, doi:10.3390/ijerph18178967_

Round 1

Reviewer 1 Report

Comments:

The study entitled as “Is it possible to have home e-monitoring of pulmonary function in our patients with Duchenne Muscular Dystrophy in the  Covid-19 pandemic (Manuscript ID#ijerph-1341493)” seems to be interesting.  In this study, the possibility to implement routine electronical monitoring of pulmonary function at home in Duchenne muscular dystrophy (DMD) patients was evaluated using an electronical spirometry system called AioCare. Such type of studies provides the basis for future innovative research as study promotes innovative digital form of monitoring pulmonary function at home, without the risk of COVID infection. Only 14% found home spirometry difficult it means it is easy to use and can be used for other patients.

Overall, the study is good, well written and very well explained but there are few minor issues that should be addressed:

Comment 1: Major limitation of this study is small sample size. 14 out of 20 boys enrolled in the study. A 25 total of 283 measurements were performed by all patients at home for 4 weeks. Only half of the patients were able to perform measurements correctly. Sample size is very low and one cannot predict the outcomes with such small sample size.

Comment 2: Only boys aged 8-12 years enrolled. Why also did not enroll boys aged 5-7 years as measurement of pulmonary function is started from the age of 5 years.

Comment 3: Page 5:  section 3.2.1: Percent of acceptable measurements (15.5%) are very low and therefore this home spirometry method does not meet the criteria for patients with Duchenne Muscular Dystrophy.

Author Response

Thank you for your valuable comments. We revised the manuscript and presented our answers below according to the Reviver’s suggestions.

The study entitled as “Is it possible to have home e-monitoring of pulmonary function in our patients with Duchenne Muscular Dystrophy in the  Covid-19 pandemic (Manuscript ID#ijerph-1341493)” seems to be interesting.  In this study, the possibility to implement routine electronical monitoring of pulmonary function at home in Duchenne muscular dystrophy (DMD) patients was evaluated using an electronical spirometry system called AioCare. Such type of studies provides the basis for future innovative research as study promotes innovative digital form of monitoring pulmonary function at home, without the risk of COVID infection. Only 14% found home spirometry difficult it means it is easy to use and can be used for other patients.

Overall, the study is good, well written and very well explained but there are few minor issues that should be addressed:

Comment 1: Major limitation of this study is small sample size. 14 out of 20 boys enrolled in the study. A 25 total of 283 measurements were performed by all patients at home for 4 weeks. Only half of the patients were able to perform measurements correctly. Sample size is very low and one cannot predict the outcomes with such small sample size.

Answer:

We do agree with the Reviewer’s comments. Due to the size of the sample, this is a pilot study indeed. We added this fact to the title of manuscript. We hope that the publication of the article as a preliminary report will meet the interest of the scientific community and will encourage further research in various scientist centers.

Comment 2: Only boys aged 8-12 years enrolled. Why also did not enroll boys aged 5-7 years as measurement of pulmonary function is started from the age of 5 years.

Answer:

We have included children who can do correctly intense maneuvers during spirometry. In our experience, children between the ages of 5 and 7 have difficulty performing the test. Therefore, the results of spirometry would be unreliable.

Comment 3: Page 5:  section 3.2.1: Percent of acceptable measurements (15.5%) are very low and therefore this home spirometry method does not meet the criteria for patients with Duchenne Muscular Dystrophy.

The acceptability parameter in spirometers is usually set for adult patients at the factory. In addition, a characteristic feature of Duchenne disease is its progressive course and progressive weakening of the respiratory muscles and even deformation of the chest. It is difficult for such patients to obtain the parameter of correctness and acceptability. To date, there is no device for measuring pulmonary function dedicated specifically to patients with DMD, which is why we have attempted to examine a home spirometer.

Reviewer 2 Report

Overall Summary. The title of this manuscript is “Is it possible to have home e-monitoring of pulmonary function in our patients with Duchenne Muscular Dystrophy in the Covid-19 pandemic?”  I found this is an interesting paper addressing an important health issue for children with DMD during the pandemic, which is timely and highly clinically relevant. The major limitations include small sample size (only 14/20 finished the study) and a short study period (1 month).   

Title

Please indicate this is a pilot study in the title.

Abstract

Please specify the study design in the abstract.

Introduction

Generally, the introduction is concise and well-written. FVC was introduced, but the clinical meaning of FEV1, PEF can also be added.

Methods

Line 80- is the observational study prospective or retrospective?

Line 108 – It is unclear how and when patients performed spirometry at the hospital. Do participants also have office visits while they are using AioCare at home? Which results were compared to home spirometry?

Line160 -extra space should be deleted

Results

Line 197 – Please specify “a modest and positive correlation”

Line 204- This sentence is confusing - Table 2 only has information for home spirometry.

Table 3 – no measurement of errors was included for Hospital spirometry

Table 5 – Because the sample size is very small, is the statistical power adequate  for performing linear regressions?

Figure 3 is unnecessary, it can be described in the context.

Is there any information on COVID-19 infection status among the participants?

Discussion

More limitations should be considered. The authors only mentioned one limitation of this study, which is small sample size. But there are more, such as generalizability (all participants are white boys and taking steroids), short study period, selection bias (only 14/20 finished the study), etc.

Author Response

Thank you for your valuable comments. We revised the manuscript and presented our answers below according to the Reviver’s suggestions.

Overall Summary. The title of this manuscript is “Is it possible to have home e-monitoring of pulmonary function in our patients with Duchenne Muscular Dystrophy in the Covid-19 pandemic?”  I found this is an interesting paper addressing an important health issue for children with DMD during the pandemic, which is timely and highly clinically relevant. The major limitations include small sample size (only 14/20 finished the study) and a short study period (1 month).  

 Title

Please indicate this is a pilot study in the title.

We do agree with the Reviewer’s comments. Due to the size of the sample, this is a pilot study indeed. We added this fact to the title of manuscript.

 Abstract

Please specify the study design in the abstract.

We added the study designed in the abstract.

Introduction

Generally, the introduction is concise and well-written. FVC was introduced, but the clinical meaning of FEV1, PEF can also be added.

Thank you for this suggestion - we added information about FEV1 and PEF into the  introduction.  

Line 80- is the observational study prospective or retrospective?

We added – a prospective study.

Line 108 – It is unclear how and when patients performed spirometry at the hospital. Do participants also have office visits while they are using AioCare at home? Which results were compared to home spirometry?

We added, that patients performed spirometry at the hospital during the first visit. Participants have no office visits while they are using AioCare at home due to pandemic time.

The results of spirometry performed during the first visit were compared to home spirometry.

Line160 -extra space should be deleted

We corrected.

Results

Line 197 – Please specify “a modest and positive correlation”

We added

Line 204- This sentence is confusing - Table 2 only has information for home spirometry.

We changed it according to the Reviewer information.

Table 3 – no measurement of errors was included for Hospital spirometry

We added

Table 5 – Because the sample size is very small, is the statistical power adequate  for performing linear regressions?

We described in “Statistical Analysis” : The goodness of fit of the model, based on Akaike’s information criterion (AIC), was compared between the four models. The number of independent variables was reduced using backward, stepwise regression models. This method allows for a decrease in the variables from the model which do not contribute to the explanation of the variability of the dependent variable which reduces the complexity of the model (which was necessary due to the limited sample size).

Of course, the results in such a small group have to be interpreted carefully

Figure 3 is unnecessary, it can be described in the context.

We added figure to better illustrate the text

Is there any information on COVID-19 infection status among the participants?

We added information about COVID 19 infection status.  All of participants presented a negative PCR-COVID test 24-48 hours before visit. Caregivers of children declared that children have not had COVID infection so far.

Discussion

More limitations should be considered. The authors only mentioned one limitation of this study, which is small sample size. But there are more, such as generalizability (all participants are white boys and taking steroids), short study period, selection bias (only 14/20 finished the study), etc.

We added to the discussion more limitations according to the Reviewer’s suggestions.